# Structure and Properties of Single-Layer MoS_2_ for Nano-Photoelectric Devices

**DOI:** 10.3390/ma12020198

**Published:** 2019-01-09

**Authors:** Jiaying Jian, Honglong Chang, Tao Xu

**Affiliations:** 1School of Mechanical Engineering, Northwestern Polytechnical University, Xi’an 710072, China; jianjiaying@mail.nwpu.edu.cn; 2School of Materials Science and Chemical Engineering, Xi’an Technological University, Xi’an 710021, China; xutaobuaa@163.com

**Keywords:** single-layer MoS_2_, chemical vapor deposition, heating method, heating temperature

## Abstract

To meet the need for preparing high-performance nano-optoelectronic devices based on single-layer MoS_2_, the effects of the heating method (one-step or two-step heating) and the temperature of the MoO_3_ source on the morphology, size, structure, and layers of an MoS_2_ crystal grown on a sapphire substrate using chemical vapor deposition are studied in this paper. The results show that MoS_2_ prepared by two-step heating (the heating of the S source starts when the temperature of the MoO_3_ source rises to 837 K) is superior over that of one-step heating (MoO_3_ and S are heated at the same time). One-step heating tends to form a mixture of MoO_2_ and MoS_2_. Neither too low nor too high of a heating temperature of MoO_3_ source is conducive to the formation of MoS_2_. When the temperature of MoO_3_ source is in the range of 1073 K to 1098 K, the size of MoS_2_ increases with the rise in temperature. A uniform large-sized triangle with a side length of 100 μm is obtained when the heating temperature of MoO_3_ is 1098 K. The triangular MoS_2_ crystals grown by the two-step heating method have a single-layer structure.

## 1. Introduction

As the first two-dimensional material discovered, graphene has shown excellent performance in electrical, optical, thermal, and mechanical aspects [1,2,3,4,5]. However, graphene is a semiconductor without a bandgap, which limits its development in optoelectronic devices. Unlike graphene, bulk, multi-layer, and single-layer MoS_2_ materials have bandgaps [6]. Bulk MoS_2_ is a semiconductor with an indirect bandgap of 1.2 eV. When the thickness of a MoS_2_ film reaches the single-layer level, the bandgap will eventually increase by at least 0.6 eV, due to the two-dimensional confinement effect. The reduction in thickness eventually causes the bandgap of the single-layer MoS_2_ film to change from an indirect bandgap to a direct one. This change in bandgap results in photoluminescence of high brightness in single-layer MoS_2_ [7]. In addition, single-layer MoS_2_ has prominent electrical performance, and the transport calculation of the nonequilibrium Green’s function shows that the switching ratio of MoS_2_ can reach up to 10^10^ [8]. Due to the superior electrical and optical performance of single-layer MoS_2_, it has potential application value in the electronic devices and photoelectronic fields [9,10,11,12,13].

MoS_2_ has a hexagonal layered crystal structure. A single-layer MoS_2_ is composed of three atomic layers, wherein the upper and lower layers are hexagonal planes composed of sulfur atoms, and the middle metal molybdenum atom layer separates the two layers of sulfur atoms. The common preparation methods for single-layer MoS_2_ include micromechanical exfoliated, intercalation of lithium ions, liquid phase ultrasonography, and the chemical vapor deposition method [14,15,16,17]. The micromechanical exfoliated method is simple and quick, and the exfoliated product is mostly single-layer MoS_2_ with high carrier mobility. Its disadvantages are that the number of exfoliated layers cannot be controlled, and industrial production is hard to realize. The intercalation of lithium ions method can prepare high-quality single-layer MoS_2_ materials through controlling the lithium ion insertion and exfoliated process by an electrochemical lithium battery device, which has the advantages of large size and high quality, but with a complicated operation process, high equipment requirements, and low exfoliated efficiency. The liquid phase ultrasonography method can prepare a single-layer MoS_2_ nanosheet. Since this method is not sensitive to water and air, it is suitable for mass production, and the obtained sheet can be easily assembled into a film. However, it is difficult to control its exfoliated degree, leaving a low concentration of the nanosheet solution after cleavage. Its disadvantage is that the ultrasonic power has great influence on the formation of the nanosheet.

The chemical vapor deposition (CVD) method uses sulfur vapor to chemically react with molybdenum oxide vapor to form MoS_2_ [18,19]. The single-layer MoS_2_ prepared by this method has excellent optical and electrical performance, and can be applied to devices like new-type resonators and transistors based on two-dimensional materials [20,21]. With a simple preparation process, the CVD method can control the structure, morphology, and size of the MoS_2_ film by controlling the reaction temperature, time, and gas flow rate. Single-layer MoS_2_ films with a size of 20 to 80 μm have been grown by CVD. However, whether larger-sized single-layer MoS_2_ films can be prepared by CVD remains unsolved.

In this paper, the effects of heating methods (one-step and two-step heating) and heating temperature of the MoO_3_ source on the morphology, size, structure, and layer number of MoS_2_ grown on a sapphire substrate by sulfurized MoO_3_ via the CVD method are investigated. 

## 2. Materials and Methods

### 2.1. Preparation of MoS_2_ Film

#### 2.1.1. Treatment of the Sapphire Substrate

A sapphire plate was chosen as the substrate to prepare an MoS_2_ crystal by vapor deposition. The sapphire substrate was cleaned in a clean room before the growth experiment. The cleaning of the sapphire plate includes two steps: the substrate was first sonicated in acetone, ethanol, and deionized water for 10 min, respectively, and then was dried with a nitrogen gas gun.

#### 2.1.2. Growth of MoS_2_ by Vapor Deposition

A chemical vapor deposition double temperature tube furnace, as shown in Figure 1, was adopted as the growth device for MoS_2_. In the experiment, the S powder was first placed in the low temperature zone on the left side of the tube furnace tube, the MoO_3_ powder was placed in the high temperature zone in the middle of the furnace tube, and the sapphire substrate was placed flat on the right side of the furnace tube. The purities of the S powder and MoO_3_ powder were 99.5%. The sulfur source in the low temperature zone was 15 cm away from the molybdenum source in the high temperature zone, and the molybdenum source was 5 cm away from the sapphire substrate.

After the tube furnace was vacuumed to 10 Torr with a mechanical pump, it was filled with high-purity argon gas at a flow rate of 70 sccm for 30 min. The evacuation–fill processes were repeated three times to vent the air and impurities in the tube furnace. After that, high-purity argon gas was introduced into the tube furnace, and when the pressure reached 1.2 atm, the exhaust valve was opened and adjusted to make the pressure in the furnace greater than 1 atm. Then, the furnace was heated to a setting temperature and held for a certain time. In the heating process, argon gas was continuously introduced into the double temperature tube furnace at a rate of 70 sccm. As a shielding gas, argon gas was also the medium to transmit the sulfur vapor from the low temperature zone to the high temperature zone. At a high temperature, the molybdenum oxide vapor was reduced by the sulfur vapor. The produced gas compound further reacted with sulfur vapor to generate MoS_2_, which was deposited onto the sapphire substrate to form MoS_2_ crystal. Since the gas volume in the tube furnace shrank with the temperature decrease during the cooling process, which decreased the pressure inside the tube, argon gas was continuously supplied until the temperature was completely lowered to room temperature to prevent outside air from entering the tube furnace.

### 2.2. Characterization and Test Methods

An SEM (scanning electron microscope; FEI-QUANTA-400, Thermo Fisher, Hillsboro, OR, USA) was used to observe the morphology of the crystal sample grown by chemical vapor deposition. The acceleration voltage of the SEM was 20 kV. For the sample grown by chemical vapor deposition, its morphology can be directly observed by the SEM. The layer number of MoS_2_ was roughly judged according to the contrast of color between MoS_2_ and the substrate. 

The layer number of MoS_2_ was determined by Raman spectroscopy (Wissenschaftliche Instrumente und Technologie GmbH, Ulm, Germany) (Renishaw-UK, with a light-passing efficiency of more than 30%, a spectral range of 200 to 1000 nm, a spectral resolution of 1 cm^−1^, and a spatial resolution of 0.5 μm in the lateral direction and 2 μm in the longitudinal direction) and a homemade photoluminescence spectrometer. The wavelength of Raman spectroscopy was 532 nm, and its grating groove density was 1800 L/mm. The wavelength of photoluminescence spectrometer was 700 nm, and its groove density was 150 L/mm.

## 3. Results and Discussion

### 3.1. Effect of Heating Method on the Morphology and Layer Number of MoS_2_ Crystals

The effects of the heating method (one-step and two-step heating) on the morphology and layer number of MoS_2_ crystals were investigated. The temperature-increase curve of one-step heating is shown in Figure 2a. In this experiment, the high temperature zone (MoO_3_) and the low temperature zone (S) were heated simultaneously at two constant rates, and they reached the set temperatures at the same time. 

By using SEM to observe the surface morphology of the one-step heating sample, it was found that a large number of white particles or quadrilateral crystals (the intermediate product MoO_2_) appeared on the surface of a large number of samples, as shown in Figure 3a. 

The above experimental results can be explained in terms of the chemical reaction and phase transition of the Mo–S–O system. In the experimental temperature region, the following reactions and phase transition can occur in the Mo–S–O system: 2S_(s)_ ⇌ S_2(g)_(1)
S_2(g)_ + 2O_2_ ⇌ 2SO_2(g)_(2)
MoO_3(s)_ ⇌ MoO_3(g)_ (*T* < 1068 K)(3)
MoO_3(s)_ ⇌ MoO_3(l)_ (*T* = 1068 K)(4)
MoO_3(l)_ ⇌ MoO_3(g)_ (*T* > 1068 K)(5)
MoO_3(g)_ + (*x*/2)S_2(g)_ ⇌ MoO_3−x(g)_ + (*x*/2)SO_2_(6)
MoO_3−x(g)_ ⇌ MoO_2(s)_ + (1/2 − *x*/2)O_2_(7)
MoO_3−x(g)_ + (7 − *x*)/S_2(g)_ ⇌ MoS_2(g)_ + (3 − *x*)/2SO_2_(8)
MoS_2(g)_ ⇌ MoS_2(s)_(9)

The effect of temperature on a chemical reaction can be judged according to the dependence of its standard Gibbs free energy change (∆*G*_0_) on the temperature (*T*). If the dependence curve of ∆*G*_0_ on *T* for a chemical reaction has a positive slope, as shown in Figure 4a, then the driving force of reaction decreases with rising temperature. On the contrary, the driving force of reaction increases with rising temperature when the slope of the curve is a negative, as shown in Figure 4b. 

The slope of the dependence curve of ∆*G*_0_ on *T* for a chemical reaction depends on the entropy change of the reaction (∆*S*_0_). When ∆*S*_0_ is positive, the slope is negative. In contrast, the slope is positive if ∆*S*_0_ is negative. It can be determined that the slopes of the dependence curve of ∆*G*_0_ on *T* for the reactions of Equations (1), (3), and (5) are negative, which indicates that the driving force of the reaction increases with rising temperature, whereas the dependence curves of ∆*G*_0_ on *T* for the reactions of Equations (2) and (7)–(9) are positive, which indicates that the driving force for the reaction decreases with rising temperature. Therefore, increasing temperature is favorable for the reactions of Equations (1), (3), and (5), but adverse for the reactions of Equations (2) and (7)–(9).

The formation of MoO_2_ depends on the reaction in Equation (7). The driving force of reaction for Equation (7) decreases with rising temperature. Therefore, the reaction of Equation (7) is easy to take place at low temperature. In addition, in one-step heating, S and MoO_3_ sources are heated at the same time. The reaction of S_2(g)_ and O_2_ can decrease the concentration of O_2_ in the furnace, which can promote the reaction of Equation (7) and produce MoO_2(s)_. Therefore, if the heating of S and MoO_3(s)_ starts at the same time, there are a large number of white or quadrilateral MoO_2_ crystals, as shown in Figure 2a on the surface of the sample. 

To reduce the formation of intermediate products, it is necessary to reduce the chance for MoO_3(g)_ to contact with S_2(g)_ at low temperatures. To achieve this purpose, a two-step heating method, as shown in Figure 2b, was employed. First, the MoO_3_ zone was rapidly heated to 837 K at a rate of 20 K/min, and then slowly heated to 1098 K at a rate of 5 K/min. The heating of the S zone began at the time when the temperature of the MoO_3_ zone was heated to 837 K, which can prevent the formation of MoO_2_, since there is no chance for MoO_3(g)_ and S_2(g)_ to contact at a temperature lower than 837 K.

Figure 3b shows the morphology of the two-step heating sample. It can be seen that there are no MoO_2_ white particles on the substrate, and the surface is very clean. In two-step heating, S has not yet sublimated when the temperature of MoO_3_ is lower than 837 K, which avoids the contact of MoO_3(g)_ and S_2(g)_ at low temperatures and reduces the possibility of generating the intermediate product MoO_2_. 

### 3.2. Effect of Temperature on the Morphology and Layer Number of MoS_2_ Crystals

The temperature directly affects the sublimation rate of MoO_3_ and S, thus affecting the growth of MoS_2_ crystal. To eliminate the influence of temperature on the volatilization rate and bring about a change in the S concentration in the tube furnace, the temperature of the low temperature zone is set at 453 K. The effect of the MoO_3_ temperature (1048 K, 1073 K, 1098 K, 1123 K) on the growth of a single-layer MoS_2_ crystal is studied with a molybdenum source dosage of 0.02 g, a holding time of 30 min, a gas flow rate of 100 sccm, and a distance between the carrier and the molybdenum source of 4~5.5 cm. 

Figure 5 shows the effect of MoO_3_ temperature on the morphology of a single-layer MoS_2_ crystal obtained by SEM. It can be seen that, as shown in Figure 5a, there is almost nothing on the substrate when the temperature of MoO_3_ is 1048 K. The reason for this is that at low temperatures, there is not enough MoO_3_ sublimated, and the amount of MoO_3_ participating in the chemical vapor deposition reaction is insufficient. When the temperature of MoO_3_ is 1073 K, as shown in Figure 5b, the temperature in the high temperature zone just exceeds the melting point of MoO_3_ (1068 K), which make it is possible for more MoO_3_ to be evaporated and reduced by S_2(g)_ to form MoS_2_. The size of the single-layer MoS_2_ depends on the evaporation pressure of MoO_3_. As shown in Figure 6, the evaporation pressure of MoO_3_ increases with rising temperature. When the temperature of MoO_3_ is 1073 K, the side length of the triangular MoS_2_ is approximately 30 μm. Obviously, the size of the triangular single-layer MoS_2_ on the substrate is not sufficiently large. When the temperature of MoO_3_ is 1098 K, as shown in Figure 5c, an amount of complete and independent triangular MoS_2_ crystals with a side length of approximately 100 μm appears. This results from the higher temperature increasing the evaporation rate of MoO_3_ and the amount of MoO_3_ evaporated, providing sufficient MoO_3(g)_ to react with S_2(g)_; thus, more MoS_2_ is formed and deposited on the substrate for crystal growth. From Figure 5b,c, it can be seen that the grown single crystals are regular triangles, the surface colors of the single crystals are uniform, and the comparisons with the substrates are apparent. It can be preliminarily judged that the triangle crystals are single-layer.

When the temperature of MoO_3_ is 1123 K, as shown in Figure 5d, the size of MoS_2_ does not become larger but diminishes. However, the MoS_2_ crystal number increases under this condition: a large number of regular triangular MoS_2_ crystals are contacted or stacked together to form multi-layered MoS_2_. Since the ∆*S*_0_ for Equation (5) is positive and ∆*G*_0_ decreases with rising temperatures, the driving force of the reaction for Equation (5) increases with rising temperature. Therefore, when the temperature of MoO_3_ rises to 1123 K, the evaporation pressure of MoO_3_ is much higher, which can increase the concentration of MoS_2(g)_ in the furnace. The critical size of the atomic cluster for MoS_2_ to form nucleation decreases with an increasing concentration of MoS_2(g)_. The decrease of the critical size of the atomic cluster for MoS_2_ to form nucleation can lead to the continuous formation of the MoS_2_ crystal nucleus, thus suppressing its growth. Therefore, when the temperature of MoO_3_ is 1123 K, a large number of multi-layered MoS_2_ crystals are formed.

In addition, at this higher temperature, the evaporation rate of MoO_3_ is much higher, and MoO_3(g)_ cannot be completely reduced by S_2(g)_ in time. The decomposition of the excessive MoO_3(g)_ can form MoO_2(s)_. Therefore, when the temperature of MoO_3_ is 1123 K, the deposits on the substrate are a mixture of MoS_2_ and disulfide.

### 3.3. Characterization of the MoS_2_ Structure

#### 3.3.1. Characterization by Raman Spectroscopy

Based on the Raman scattering effect, Raman spectroscopy can be used to study the structure of molecules by analyzing the scattering spectrum different from the incident light frequency. There are five vibration modes ($E_{2}^{2}$_g_, $E_{2}^{1}$_g_, E_1g_, E_1u_, and A_1g_) in MoS_2_. For the Raman spectroscopy test of MoS_2_, only $E_{2}^{1}$_g_ and A_1g_ modes are observed [22,23]. The characteristic peak position of the Raman spectrum of MoS_2_ is related to the film thickness. The Van der Waals force increases as the layer number decreases. When the layer number of the film decreases gradually, the characteristic peak corresponding to the A_1g_ vibration mode happens to red shift (shifts to the long wavelength side), and the characteristic peak corresponding to the $E_{2}^{1}$_g_ vibration mode happens to blue shift (shifts towards the blue end of the spectrum). When MoS_2_ is a bulk material, the $E_{2}^{1}$_g_ peak is located near 382 cm^−1^, and the A_1g_ peak is located near 407 cm^−1^. When the layer number of MoS_2_ is reduced to a single layer, the $E_{2}^{1}$_g_ peak shifts towards the blue end, to approximately 385 cm^−1^, and the A_1g_ peak shifts to the long wavelength side, to approximately 403 cm^−1^. The A_1g_ peak corresponding to MoS_2_ film of different layers differs from the $E_{2}^{1}$_g_ peak, and thus the layer number of MoS_2_ could be determined based on the difference between the detected A_1g_ peak and $E_{2}^{1}$_g_ peak [24]. Figure 7 shows the results of the clean sapphire substrate and the single-layer triangular MoS_2_ samples deposited on the substrate obtained by Raman spectroscopy. By comparing Figure 7a,b, it can be inferred that the peak appearing at a wavenumber of 416 cm^−1^ in Figure 7b is caused by the influence of the sapphire substrate Al_2_O_3_. The temperature in the high temperature zone of sample 1 (the sample in Figure 5b) is 1073 K, and the temperature in the high temperature zone of the sample 2 (the sample in Figure 5c) is 1098 K. It can be seen that the $E_{2}^{1}$_g_ peak of the single-layer triangular MoS_2_ in the two samples is approximately 385 cm^−1^, and the A_1g_ peak is approximately 403 cm^−1^. The wavenumber difference is 19.2 cm^−1^ and 20.7 cm^−1^, respectively. This indicates that the deposits on the sapphire substrate of sample 1 and sample 2 are single-layer MoS_2_, which is consistent with the predicted results of SEM.

#### 3.3.2. Characterization of Photoluminescence Spectra

In bulk MoS_2_, no photoluminescence is observed. When it is gradually thinned to a few layers, or even a single layer, the band gap of MoS_2_ changes to a direct band gap, and the fluorescence effect is obviously enhanced, with two characteristic peaks appearing near 1.8 eV and 2.0 eV. The characteristic peak corresponding to 1.8 eV is the direct band gap of MoS_2_, and the characteristic peak corresponding to 2.0 eV is the energy band splitting caused by the spin orbit coupling of MoS_2_ [7,25]. The photoluminescence spectrum can be used as an effective means of characterizing the layer number of MoS_2_. Figure 8 shows the results of the clean sapphire substrate and the single-layer triangular MoS_2_ samples deposited on the substrate obtained by photoluminescence spectra. By comparing Figure 8a,b, it can be believed that the peak appearing at a wavelength of 692.7 nm in Figure 8b is caused by the influence of the sapphire substrate Al_2_O_3_. Sample 1 (the sample in Figure 5b) shows a strong characteristic peak at the wavelength of λ_1_ = 665.8 nm, and sample 2 (the sample in Figure 5c) shows a strong characteristic peak at λ_2_ = 671.4 nm. According to the Planck–Einstein relation, it can be determined that the corresponding transition energy level E = 1.86 eV at the luminescence peak λ_1_ = 665.8 nm, and the corresponding transition energy level E = 1.85 eV at the luminescence peak λ_2_ = 671.4 nm, which is in accordance with the band gap width of single layer MoS_2_. It can be seen that both the characteristic peaks around 670 nm of two samples are intrinsic transition peaks, further verifying that the MoS_2_ crystals deposited on the substrate in Figure 5b,c are of single-layer structures. By comparing the fluorescence efficiency of two samples at the characteristic peaks around 670 nm, it can be seen that the fluorescence efficiency of sample 1 is higher than that of sample 2.

## 4. Conclusions

(1) The two-step heating method to prepare MoS_2_ is superior to the one-step heating method. Mixed compounds of MoO_2_ and MoS_2_ are easily formed by the one-step heating method. The two-step heating method can avoid the contact of S_2(g)_ with MoO_3(g)_ and the formation of MoO_2_ at low temperatures.

(2) The growth of MoS_2_ crystal depends on the temperature. Neither too low nor too high of a heating temperature of MoO_3_ is conducive to the formation of MoS_2_. In the range of 1073 K to 1098 K, the size of MoS_2_ increases with the rise in temperature. A uniform large-sized triangle with a side length of 100 μm is obtained when the heating temperature of MoO_3_ is 1098 K.

(3) The test results of SEM, Raman spectroscopy, and photoluminescence spectroscopy show that the triangular MoS_2_ crystals grown by the two-step heating method are single-layer structures. 

## Figures and Tables

**Figure 1 materials-12-00198-f001:**
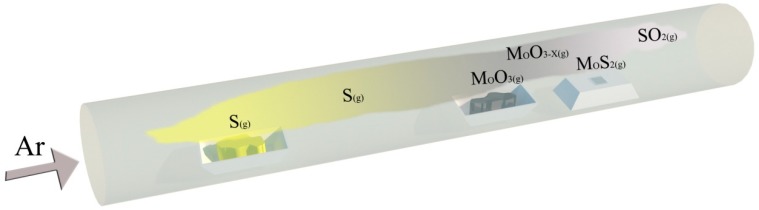
Schematic of the preparation of MoS_2_ by chemical vapor deposition (CVD).

**Figure 2 materials-12-00198-f002:**
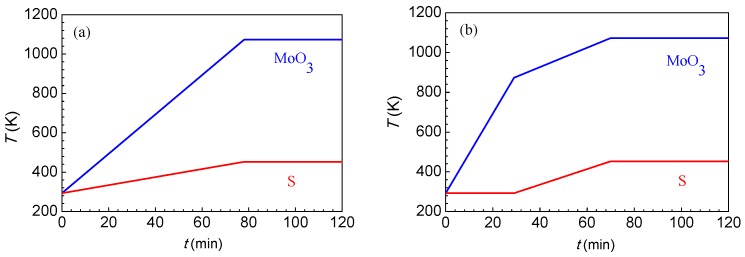
Heating methods to prepare MoS_2_: (**a**) one-step heating and (**b**) two-step heating.

**Figure 3 materials-12-00198-f003:**
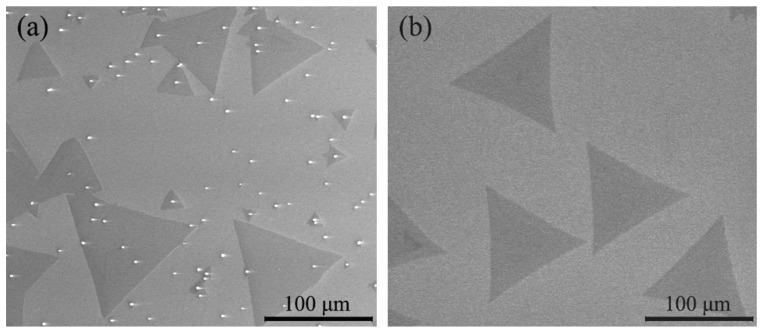
Scanning electron microscope (SEM) images of MoS_2_ prepared by using the two heating methods: (**a**) one-step heating and (**b**) two-step heating.

**Figure 4 materials-12-00198-f004:**
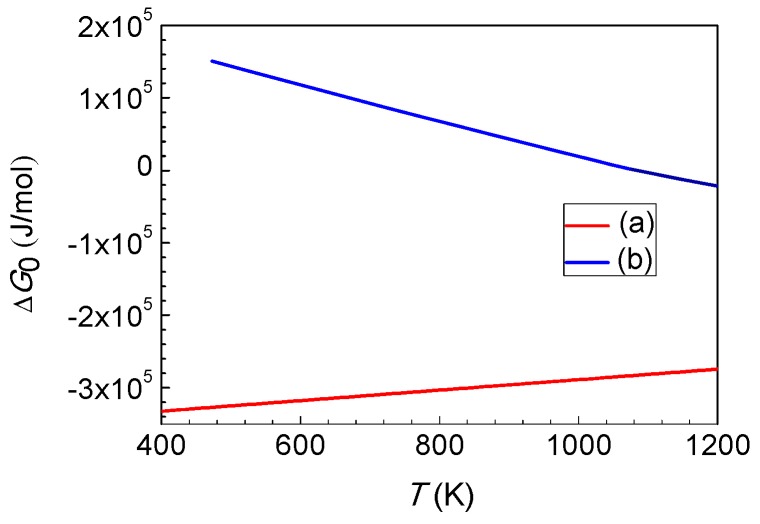
Dependence curve of the standard Gibbs free energy change (∆*G*_0_) on temperature (*T*) for the chemical reaction. (**a**) Positive slope (∆*S*_0_ < 0); (**b**) negative slope (∆*S*_0_ > 0).

**Figure 5 materials-12-00198-f005:**
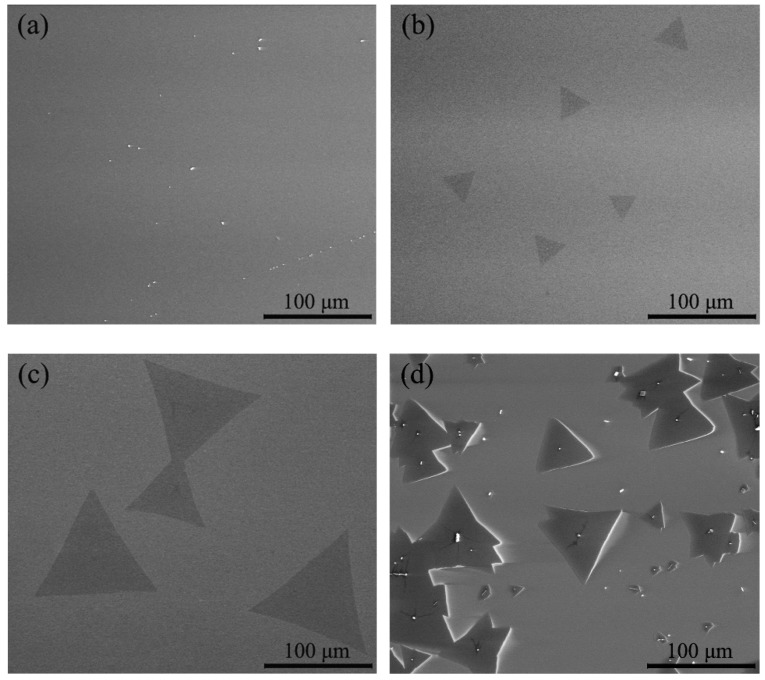
SEM images of MoS_2_ prepared from different temperatures: (**a**) 1048 K, (**b**) 1073 K, (**c**) 1098 K, (**d**) 1123 K.

**Figure 6 materials-12-00198-f006:**
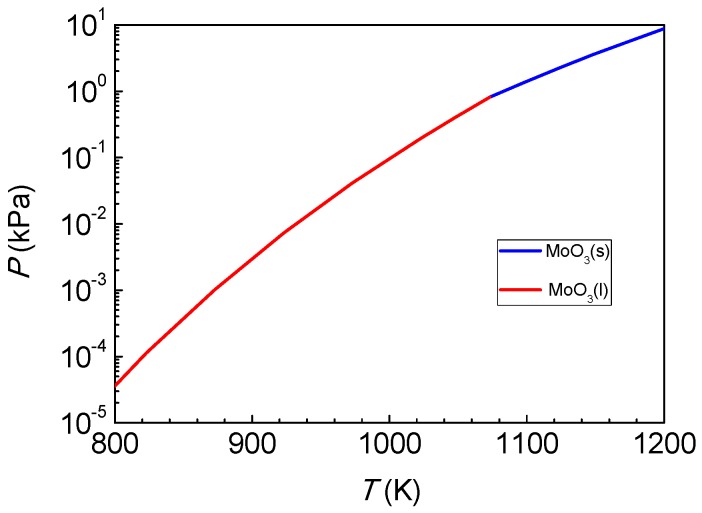
Dependence of the evaporation pressure *P* on the temperature *T* for MoO_3_.

**Figure 7 materials-12-00198-f007:**
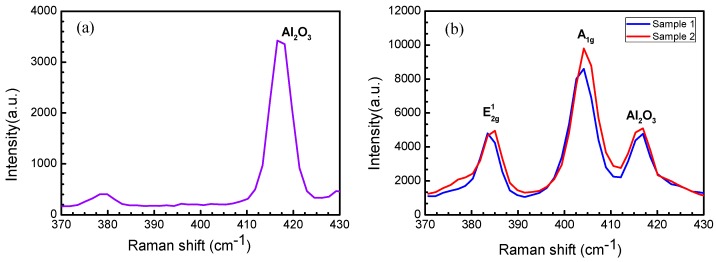
Raman spectra. (**a**) Clean sapphire substrate and (**b**) single-layer triangular MoS_2_ samples deposited on the substrate. Sample 1: the temperature in the high temperature zone is 1073 K; Sample 2: the temperature in the high temperature zone is 1098 K.

**Figure 8 materials-12-00198-f008:**
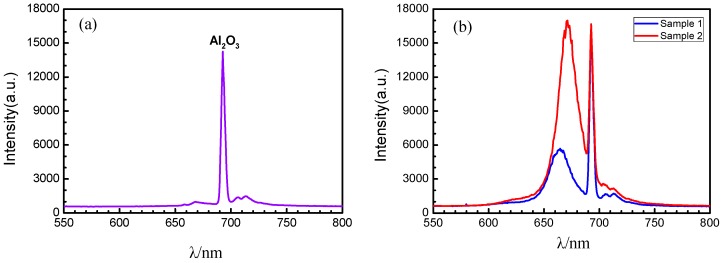
Photoluminescence spectra. (**a**) Clean sapphire substrate and (**b**) single-layer triangular MoS_2_ samples deposited on the substrate. Sample 1: the temperature in the high temperature zone is 1073 K; Sample 2: the temperature in the high temperature zone is 1098 K.

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
