# Peer review of "Structure and Properties of Single-Layer MoS2 for Nano-Photoelectric Devices"

_materials, 2019, doi:10.3390/ma12020198_

Round 1

Reviewer 1 Report

The authors report the control of a single-layer MoS2 nanocrystal on sapphire substrate by the temperature control experiments.  I wonder the manuscript is suitable for Materials because the descriptions are poor but the level may increase after the correction as shown below.  I strongly recommend the manuscript should be modified appropriately as an academic paper for the publication level.

1.     Background of the research and the motivation

Fabrication of monolayer MoS2 have been reported in many literature by CVD method. The size control experiments have also been reported by some research groups. What is the originality of the manuscript?

2.     Reference format and citation

There are many errors in the reference list. Abbreviation, capitalize, pages and so on. In addition, many citations were missing. The authors should cite the appropriate literature in the main body.

3.     Main body

1)     There are some grammar errors in the manuscript. Please modify them.

2)     What are the meaning of the equal sign in the chemical reaction?  The reaction (1)-(9) contain reversible (phase transition) reactions and irreversible reactions.  They should be clear.

3)     [p. 3, L. 98] What is the acceleration voltage of the SEM? The information is required, because the authors roughly estimated the number of the layers by the SEM image. But the film thickness was not described in the manuscript...

4)     [p. 3, L. 103] The unit of the Raman spectral range is really not nm but wavenumber (cm-1)?  In addition, the incident wavelength of the Raman spectra should be written in a laser Raman system.

5)     [p. 4, L. 134] The definition of “standard” is the 298 K temperature and 1 bar pressure. The standard Gibbs free energy is not dependent on temperature! The experimental condition of CVD is 10 Torr and very high temperature. Therefore, ”standard” should be removed and the meaning of the ΔG should be clarify in the manuscript.

6)     [p. 4, L. 159] The unit of the temperature should be uniformed to Kelvin. Moreover, the unit “T/min” should be K/min.

7)     [p. 7, L. 216] Why all characteristic modes of the Raman signal were not observed without E2g1 and A1g?

8)     [p. 7, L. 218] The authors showed the peaks of the Raman spectrum are related to the film thickness. Why do the peaks depend on the film thickness?  The literature should be cited in the manuscript.

9)     [p. 7, L. 233] What is the literature in “According to the literature”?

10) [p. 7, L. 235] What is the definition of “Sample 1” and “Sample 2”?

11)  In the end, what was the thickness of MoS2 in each substrate? And how was the consistency of the contrast of SEM images and the peak of the Raman spectra?

12) [p. 8, L. 249] The authors should show the literature.

13) [p. 8, L. 253] What is the excitation wavelength of the emission spectra?  And what did the difference of the emission peak intensities at 1.8 and 2.0 eV originate from by comparison with two samples?

Author Response

1. Background of the research and the motivation Fabrication of monolayer MoS2 have been reported in many literature by CVD method. The size control experiments have also been reported by some research groups. What is the originality of the manuscript? Response: This paper reports a new method of fabrication of large size monolayer MoS2. Large size monolayer MoS2 can be easily obtained by this method. The repeatability and stability of this method is better than those of the methods reported in previous literatures. In addition, the thermodynamic mechanism for fabrication of large size monolayer MoS2 by this method is revealed. 2. Reference format and citation There are many errors in the reference list. Abbreviation, capitalize, pages and so on. In addition, many citations were missing. The authors should cite the appropriate literature in the main body. Response: We have made correction according to the Reviewer’s comments. 3. Main body 1) There are some grammar errors in the manuscript. Please modify them. Response: We are sorry for the errors and have modified them. 2) What are the meaning of the equal sign in the chemical reaction?  The reaction (1)-(9) contain reversible (phase transition) reactions and irreversible reactions.  They should be clear. Response: All the reactions in this paper are reversible. The equal signs in equations (1)-(9) have been replaced by reversible reaction signs. 3) [p. 3, L. 98] What is the acceleration voltage of the SEM? The information is required, because the authors roughly estimated the number of the layers by the SEM image. But the film thickness was not described in the manuscript... Response: The acceleration voltage of the SEM has been added. We have added the description of the number of layers by the SEM image in the manuscript. 4) [p. 3, L. 103] The unit of the Raman spectral range is really not nm but wavenumber (cm-1)?  In addition, the incident wavelength of the Raman spectra should be written in a laser Raman system. Response: Generally, the transverse axis of Raman spectral is the wavenumber (cm-1). The wavenumber can also be translated into the wavelength. The incident wavelength of the Raman spectra has been added. 5) [p. 4, L. 134] The definition of “standard” is the 298 K temperature and 1 bar pressure. The standard Gibbs free energy is not dependent on temperature! The experimental condition of CVD is 10 Torr and very high temperature. Therefore, ”standard” should be removed and the meaning of the ΔG should be clarify in the manuscript. Response: The standard Gibbs free energy change of reaction is the standard free energy change of the reaction under the standard state. In international standards, the standard state for a gas phase is specified as the state when the gas partial pressure is 1 atm; the standard state for a solute in an alloy is specified as the state when the mass percentage concentration of the solute is 1 %, and the standard state for a condensed phase is specified as the state when the mole fraction of the condensed phase is 1. When the reaction is to form a compound, the standard Gibbs free energy change of reaction is called the standard Gibbs free energy of formation of compound. The standard free energy change of reaction can be calculated according to the standard Gibbs free energy of formation. For example: MoO3(s) = MoO3(g)             (1)   Mo(s) +O2(g) = MoO3(s)      (2)   Mo(s) +O2(g)= MoO3(g),          (3)   If the standard Gibbs free energy change of reaction for equation (1), the standard Gibbs free energies of formation for equations (2) and (3) are expressed as G0(1), G0(2) and G0(3), respectively, G0(1) can be determined as follow: G0(1)= G0(3)-G0(2) The standard Gibbs free energy of formation for compound (it can be taken from Handbook of Chemistry) is the function of temperature, so the standard Gibbs free energy change of reaction is the function of temperature. 6) [p. 4, L. 159] The unit of the temperature should be uniformed to Kelvin. Moreover, the unit “T/min” should be K/min. Response: We are sorry for our misspellings and we have made corrections. 7) [p. 7, L. 216] Why all characteristic modes of the Raman signal were not observed without E1 2g and A1g? Response: We have cited two literatures to explain why all characteristic modes of the Raman signal were not observed without E1 2g and A1g. 8) [p. 7, L. 218] The authors showed the peaks of the Raman spectrum are related to the film thickness. Why do the peaks depend on the film thickness? The literature should be cited in the manuscript. Response: The literature has been cited. 9) [p. 7, L. 233] What is the literature in “According to the literature”? Response: We have added the results of a clean sapphire. 10) [p. 7, L. 235] What is the definition of “Sample 1” and “Sample 2”? Response: We have cited “Sample 1” and “Sample 2” in the manuscript. 11)  In the end, what was the thickness of MoS2 in each substrate? And how was the consistency of the contrast of SEM images and the peak of the Raman spectra? Response: By SEM, we can see whether the deposits on the substrates are single triangles (Figure 5b and 5c) or stacked triangles (Figure 5d). Single triangular sediment may be single-layer or multilayer. But in general, the triangular sediments with uniform colour are mostly single-layer. Therefore, SEM can only preliminarily determine whether the sediment is single-layer or not. Raman spectroscopy and photoluminescence spectroscopy are mature characterization methods determining the number of layers and whether the sediment is single-layer or not. 12) [p. 8, L. 249] The authors should show the literature. Response: The literature has been cited. 13) [p. 8, L. 253] What is the excitation wavelength of the emission spectra?  And what did the difference of the emission peak intensities at 1.8 and 2.0 eV originate from by comparison with two samples? Response: The excitation wavelength of the emission spectra has been added. We have compared the difference of the characteristic peak between the two samples.

Reviewer 2 Report

The manuscript by Jiaying Jian et al. discusses the effects of two heating methods and the temperature of MoO3 source on the morphology, size, structure and layers of a MoS2 crystal grown on sapphire substrate.

Few remarks:

1) pag. 3 - section 2.2

“For the sample grown by chemical vapor deposition, its morphology can be directly observed by the SEM. The layer number of MoS2 was roughly judged according to the contrast of color between MoS2 and the substrate. 

The layer number of MoS2 was determined by Raman spectroscopy (Renishaw-UK, with a light-passing efficiency of more than 30%, a spectral range of 200 to 1000 nm, a spectral resolution of 1 cm-1, and a spatial resolution of 0.5 um in the lateral direction and 2 um in the longitudinal direction) and a homemade photoluminescence spectrometer.’

How the layer number of MoS2 was really evaluated?

2) pag.4 - section 3.1 

“as shown in Figure 4(a), the driving force of reaction decreases with increasing temperature, whereas the driving force of reaction increases with increasing temperature if the slope of the curve is a negative, as shown in Figure 4(b)” 

(a) and (b) refer to the curves in Figure 4. Please modify the text.

3) pag. 8 - raw 246/247

What does it mean the “electronic Ford conversion”?

4) pag. 8 - Figure 8.

What are the sample 1 and the sample 2? Why they are different?

A revision of the language is also strongly suggested.

Author Response

1) pag. 3 - section 2.2 “For the sample grown by chemical vapor deposition, its morphology can be directly observed by the SEM. The layer number of MoS2 was roughly judged according to the contrast of color between MoS2 and the substrate. The layer number of MoS2 was determined by Raman spectroscopy (Renishaw-UK, with a light-passing efficiency of more than 30%, a spectral range of 200 to 1000 nm, a spectral resolution of 1 cm-1, and a spatial resolution of 0.5 um in the lateral direction and 2 um in the longitudinal direction) and a homemade photoluminescence spectrometer.’ How the layer number of MoS2 was really evaluated? Response: We determined the layer number in part 3.3(Characterization of MoS2 Structure, Characterization by Raman Spectroscopy). 2) pag.4 - section 3.1 “as shown in Figure 4(a), the driving force of reaction decreases with increasing temperature, whereas the driving force of reaction increases with increasing temperature if the slope of the curve is a negative, as shown in Figure 4(b)” (a) and (b) refer to the curves in Figure 4. Please modify the text. Response: We have modified in the manuscript. 3) pag. 8 - raw 246/247 What does it mean the “electronic Ford conversion”? Response: We are sorry for our incorrect writing and we have made a correction. 4) pag. 8 - Figure 8. What are the sample 1 and the sample 2? Why they are different? Response: We have cited “Sample 1” and “Sample 2” in the manuscript.

Round 2

Reviewer 1 Report

The authors have modified their manuscript appropriately. Therefore, I think the quality of the manuscript will reach Materials after the brief check of the English language and the reference section which has still some misdescription e.g. on capitalization and abbreviation.